# On the quantum simulation of complex networks

**Duarte Magano[1,2], João P. Moutinho[1,2] and Bruno Coutinho[1]**

**1** Instituto de Telecomunicações, Physics of Information and Quantum Technologies Group,
Av. Rovisco Pais 1, 1049-001, Lisbon, Portugal
**2** Instituto Superior Técnico, Universidade de Lisboa,
Av. Rovisco Pais, 1049-001 Lisboa, Portugal

## Abstract

Quantum walks provide a natural framework to approach graph problems with quantum computers, exhibiting speedups over their classical counterparts for tasks such as the search for marked nodes or the prediction of missing links. Continuous-time quantum walk algorithms assume that we can simulate the dynamics of quantum systems where the Hamiltonian is given by the adjacency matrix of the graph. It is known that such can be simulated efficiently if the underlying graph is row-sparse and efficiently row-computable. While this is sufficient for many applications, it limits the applicability for this class of algorithms to study real world complex networks, which, among other properties, are characterized by the existence of a few densely connected nodes, called hubs. In other words, complex networks are typically not row-sparse, even though the average connectivity over all nodes can be very small. In this work, we extend the state-of-the-art results on quantum simulation to graphs that contain a small number of hubs, but that are otherwise sparse. Hopefully, our results may lead to new applications of quantum computing to network science.

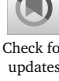

# 1  Introduction

A network is a set of distinct interacting elements, often represented as a graph. Such a general description can be applied to many complex systems from diverse fields such as biology, medicine, technology, finance, sociology and others [1, 2]. The modeling of such networks has proven to be of practical interest in understanding these complex systems. To give a few examples, today's disease spreading algorithms use social network data to improve their accuracy [3, 4], and the human interactome (the set of all protein-interactions in the human cell) is a valuable tool in the discovery of new cancer genes [5] and drug combinations [6]. The computational cost of analyzing complex networks is constantly growing, as more data becomes available and finer details are modeled [7, 8]. Considering the number of elements alone, complex networks with up to billions of nodes appear in different fields, for example, the neuronal network in the human brain [9] or the World Wide Web [10]. Overall, it is essencial to develop novel and more efficient algorithms for complex network analysis.

With the first working quantum processors on the horizon, it is fair to wonder what may be their applications to network science. Quantum computers are already known to have an advantage over classical ones for a number of graph problems. During the past twenty-five years quantum algorithms have been put forward for traversing graphs [11, 12], computing NAND trees [13], finding marked nodes in graphs [14, 15], solving optimization problems [16, 17], and predicting missing links in a network [18, 19]. All the aforementioned algorithms are based on continuous-time quantum walks. That is to say that they rely on the ability to transform quantum states as

$$|\psi_0\rangle \rightarrow |\psi(t)\rangle := e^{-iAt} |\psi_0\rangle \,, \tag{1}$$

where $A$ is the adjacency matrix of the underlying graph. For example, Ref. [18] implements $e^{iAt} \pm e^{-iAt}$ to encode weighted sums of even/odd powers of the adjacency matrix, which are then used as path-based prediction scores for new links. As another example, Ref. [15] performs the time evolution $e^{-iAt}$ for an appropriately chosen random time $t$, preparing a mixed state that is later measured to find marked nodes.

Why should we expect to be able to implement the transformation (1)? Quantum walk algorithms often treat the unitary $e^{-iAt}$ as if it was a native operation of the quantum computer. But one must consider the actual cost of implementing it on a universal quantum computer. Childs *et al* [12] showed how to simulate a quantum walk on a graph of glued trees (provided with a proper edge coloring). Modern algorithms on Hamiltonian simulation can implement

a quantum walk on any graph whose adjacency matrix is row-sparse[1] and efficiently row-computable [20–22]. Concretely, consider an input model where we can query the entries of the matrix $A$ and of the adjacency list of the graph. Then, if every node in the graph has degree at most $d$ we can approximate $e^{-iAt}$ up to precision $\epsilon$ using

$$\mathcal{O}(dt + \log(1/\epsilon)) \tag{2}$$

queries [22].

If we want to run quantum walks on complex networks, these algorithms become inefficient [19]. Indeed, a defining characteristic of complex networks, found in networks with completely different origins, is the presence of a heavy-tailed distribution of the degrees of the nodes (such as a power-law, or a log-normal distribution). That is, the tail of the degrees' distribution is not exponentially bounded, implying a non-negligible probability of observing *hubs*, nodes with a much larger number of connections than the rest of the network. Clearly, a network with hubs is not sparse. In these cases, the $d$ factor in expression (2) cannot be described as a polylogarithmic function of the size of the network and we lose the notion of efficiency in the simulation of the quantum walk.

In this work, we propose a model of hub-sparse networks, where we add a few densely connected hub-nodes to an otherwise sparse network, and show that hub-sparse networks can be efficiently simulated on a quantum computer. While the hub-sparse model is an oversimplification of real networks, we hope that our results may constitute a relevant first step towards understanding whether quantum computers can provide an advantage for simulating complex networks.

The paper is organized as follows. In section 2, we summarize the main ideas of our work: in section 2.1 we introduce the hub-sparse model, in section 2.2 we lay down the input model of our problem, and in section 2.3 we state the main theorem and provide a high-level overview of the proof. In section 3 we review some background material that may help the reader follow the technical details of our proof. The full proof of the main theorem is detailed in sections 4–6: section 4 introduces new operators that convey information about the graph and explains how they can be built from the input model, section 5 shows how to fast-forward the network of connections between hubs and regular connections, and section 6 presents the final algorithm based on the interaction picture of quantum simulation. Section 7 concludes the paper with a discussion of our results.

**Notation**    Whenever talking about graphs, we denote the number of nodes by $N$. For simplicity, we assume it to be a power of two: $N =: 2^n$. We write $[N] := \{0, 1, \dots, N-1\}$. We use $A$ to denote the adjacency matrix of the graph. The degree of a node is the number of nodes that it connects to. When we write log, we mean base 2 logarithm. We adopt the standard "big O" notation for asymptotic upper bounds. For two functions $f$ and $g$ from $\mathbb{N}$ to $\mathbb{N}$ we say that $f = \mathcal{O}(g)$ if $\exists C, x_0 > 0 : \forall x, (x > x_0 \implies f(x) < C \cdot g(x))$, and say that $f = \Omega(g)$ if $g = \mathcal{O}(f)$. Finally, we write $f(x) = \text{polylog}(x)$ if $f(x) = \mathcal{O}(\log^c(x))$ for some constant $c$.

## 2 Summary

### 2.1 Hub-sparse networks

In the context of quantum computing, "Hamiltonian simulation" refers to the following problem.

---

[1]We follow the computer science convention to define the term "sparse". We say that a network is "sparse" or "row-sparse" if the maximum degree is at most polylogarithmic on the total number of nodes. Nonetheless, it should be noted that the network science community often uses "sparse" to designate any network whose average degree is much smaller than the number of nodes.

**Definition 2.1** (Hamiltonian simulation). Given a description of a $n$-qubit Hamiltonian $H(\tau)$, an evolution time $t$, an initial state $|\psi_0\rangle$, and a precision $\epsilon > 0$, implement a unitary $U_t$ such that

$$\| U_t |\psi_0\rangle - |\psi(t)\rangle \| \leq \epsilon \,, \tag{3}$$

where $|\psi(t)\rangle$ is the solution of the differential equation

$$i\partial_\tau |\psi(\tau)\rangle = H(\tau)|\psi(\tau)\rangle \,, \tag{4}$$

at $\tau = t$ subject to $|\psi(0)\rangle = |\psi_0\rangle$. If $H$ is time-independent, then we just want to approximate $\exp(-iHt)$. We say that a Hamiltonian simulation algorithm is *efficient* if its complexity is polynomial in $n$.

An important sub-class of Hamiltonians are Hermitian matrices with binary entries, as they can be interpreted as adjacency matrices of graphs. Simulating a $N$-node graph $G$ means solving Hamiltonian simulation with the $\lceil \log(N) \rceil$-qubit Hamiltonian corresponding to the adjacency matrix of the graph.

The previous literature on quantum simulation algorithms has worked with *sparse* graphs, for which it has been shown that there are efficient Hamiltonian simulation algorithms [20, 21, 23]. A graph is said to be $d$-sparse if each node connects to at most $d$ other nodes. By sparse network we mean that the underlying graph is polylog($N$)-sparse. This does not capture the essence of complex networks due to the existence of densely connected hub-nodes.

In this work, we introduce a broader class of networks that we refer to as *hub-sparse* networks. These have a bimodal character. Most of the nodes connect only to a few other nodes, but a small fraction of nodes connects to almost the entire network. In other words, we are adding hubs to networks that are otherwise sparse. See Figure 1a for an illustration of this class of networks. We provide a rigorous definition below.

**Definition 2.2** (Hub-sparse networks). A graph $G = (V, E)$ with $N$ nodes is hub-sparse if there are $M, h, s = \mathcal{O}(\text{polylog}(N))$ and a bipartition of the nodes $\{V_h, V_r\}$ such that

1. $|V_h| = M$ (and so $|V_r| = N - M$),

2. every node in $V_h$ connects to at least $N - h$ other nodes,

3. every node in $V_r$ connects to at most $s$ other nodes.

We say that a given node is a hub if its degree is $\Omega(N - \text{polylog}(N))$, and that it is a regular node if its degree is $\mathcal{O}(\text{polylog}(N))$.

Unless otherwise stated, from now on we will always assume that we are working with hub-sparse networks. We will also assume that we know the values of $M$, $h$, and $s$, as in Definition 2.2. For simplicity, we assume that these are powers of two (although not much changes otherwise).

## 2.2 Input model

Like in the works on the simulation of sparse Hamiltonians, we assume that we can quickly recognize and locate the non-zero entries of the adjacency matrix of the graph. Specifically, denoting $A$ as the adjacency matrix of the graph, one assumes access to a quantum oracle $O_A$ acting as

$$O_A |i, j, z\rangle = |i, j, z \oplus A_{ij}\rangle \,, \tag{5}$$

with $i, j \in [N]$ and $z \in \{0, 1\}$; as well as to another oracle $O_L$ such that

$$O_L |i, l\rangle = |i, r(l, i)\rangle \,, \tag{6}$$

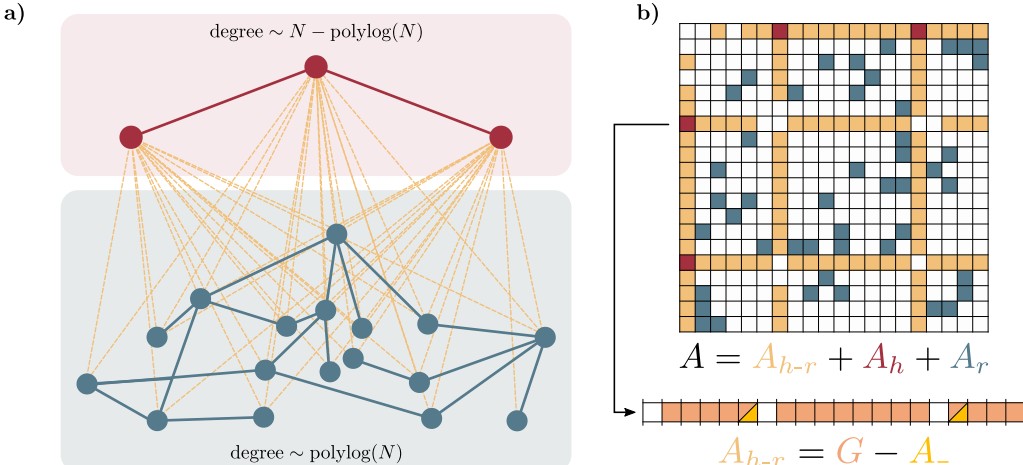

Figure 1: Hub-sparse networks. In panel $a$) we show an example of a hub-sparse network with twenty nodes. There is a small number of hubs (red) that are connected to most of the network, while the regular nodes (blue) only connect to a few other nodes. In general, a hub-sparse network contains $\text{polylog}\,N$ hub nodes with degree $\Omega(N - \text{polylog}\,N)$, the remaining nodes having degree $\text{polylog}\,N$ – Definition 2.2 In panel $b$) we plot the corresponding adjacency matrix, $A$. In our algorithm, we split $A$ into $G$, $A_-$, $A_h$, and $A_r$ (*cf.* section 2). $G$ is the matrix of all possible links between hubs and regular nodes (equation (9)). $A_-$ is the matrix of links between hubs and regular nodes that are not present in the network (equation (10)) and $A_h$ and $A_r$ are the matrices of connections between hubs and regular nodes, respectively. Overall, we can write $A = G - A_- + A_h + A_r$.

with $i \in [N]$ and $r(l,i)$ being the position of the $l$-th non-zero entry in $i$-row of $A$. If there are less than $l$ non-zero entries in the $i$-th row, we assume that $r(l,i)$ contains a flag indicating so. We also assume that we can quickly find the hubs. We express this via an oracle $O_H$ acting as

$$O_H |l\rangle = |h(l)\rangle \,, \tag{7}$$

where $h(l)$ is the $l$-th hub in the network.[2]

We also assume access to the controlled versions of the oracles $O_A$, $O_L$, and $O_H$ and their inverses.

## 2.3 Main results

Our main result is an extension of previous results on efficient Hamiltonian simulation from sparse to hub-sparse networks.

**Theorem 2.1.** *Let $A \in \{0,1\}^{N \times N}$ be the adjacency matrix of a hub-sparse network. Let $t, \epsilon > 0$. Then, there is a quantum algorithm that prepares $\exp(-iAt)$ with precision $\epsilon$ and error probability $\mathcal{O}(\epsilon)$ with*

$$\mathcal{O}\left(t\,\text{polylog}(t/\epsilon, N)\right) \tag{8}$$

*calls to the controlled and inverse versions of the oracles $O_A$, $O_L$, and $O_H$, and primitive two-qubit gates.*

---

[2]Such a unitary exists because $h(l)$ is one-to-one. We could have postulated a different oracle $O'_H$ acting as $O'_H |l\rangle |z\rangle = |l\rangle |z \oplus h(l)\rangle$. $O'_H$ can be straightforwardly simulated by $O_H$. Choosing $O_H$ or $O'_H$ for the input model does not change the main conclusions of the paper.

We now provide a high-level overview of our proof. Some of the underlying concepts (block-encodings, fast-forwarding, linear combination of unitaries, time-dependent Hamiltonian simulation) are introduced in the Preliminaries section. The full demonstration of Theorem 2.1 is detailed in Sections 4–6.

Hamiltonian simulation is an eigenvalue transformation problem. By the Quantum Singular Value Transformation Theory [24], if we can efficiently block-encode $H/\alpha$, then we can implement $\exp(-itH)$ up to constant error in $\mathcal{O}(t\alpha)$ time. A fundamental limitation of this technique is that we can never block-encode $H/\alpha$ with $\alpha < \|H\|$. The spectral norm of the adjacency matrix of a hub-sparse network scales as $\|A\| \sim \sqrt{N}$. That is, the cost associated with the block-encoding is exponential in $n = \log(N)$, and there is no hope of obtaining an efficient simulation algorithm this way.

We overcome this challenge by splitting the graph into simpler parts. Let $G$ be the matrix of every possible link between the set of hubs and the set of regular nodes for a given graph,

$$G_{ij} := \begin{cases} 1, & \text{if } (i \text{ is hub}) \text{ XOR } (j \text{ is hub}), \\ 0, & \text{otherwise}. \end{cases} \tag{9}$$

Then, the matrix of observed links between hubs and regular nodes in the graph is $A_{h\text{-}r} := G \odot A$, where $\odot$ is the Hadamard (or element-wise) product. Similarly, we can define $A_-$ as the matrix of links between hubs and regular nodes that are not in $A_{h\text{-}r}$, i.e., not observed in the graph,

$$A_- := G - A_{h\text{-}r}. \tag{10}$$

Finally, we can define $A_h$ and $A_r$ as the matrices of hub-hub and regular-regular connections, respectively, and write

$$A - A_{h\text{-}r} = A_h + A_r. \tag{11}$$

In summary, by decomposing $A$ into these various matrices, the problem that we want to solve is

$$i\partial_\tau |\psi(\tau)\rangle = \big(G - A_- + A_h + A_r\big)|\psi(\tau)\rangle, \quad \forall \tau \in [0, t]. \tag{12}$$

We refer to Figure 1b for a visualization of this splitting.

The three matrices $A_-, A_h, A_r$ are sparse, and their spectral norms are in polylog($N$). However, the norm of $G$ is $\mathcal{O}(\sqrt{N})$, and so it cannot be efficiently block-encoded. Fortunately, $G$ has enough structure that it can be fast-forwarded, i.e., the complexity of its simulation scales better than $\|Gt\|$. To do so, we show that $G$ has only two eigenvectors, say $|\Psi_\pm\rangle$, with eigenvalues different from zero, $\lambda_\pm$. Therefore,

$$e^{-iGt} = (e^{-i\lambda_+} - 1)|\Psi_+\rangle\langle\Psi_+| + (e^{-i\lambda_-} - 1)|\Psi_-\rangle\langle\Psi_-| + I. \tag{13}$$

We show that the states $|\Psi_\pm\rangle$ can be prepared efficiently, allowing the simulation of $\exp(-iGt)$ in polylog($N$) time.

Following Low and Wiebe's interaction picture simulation method [25], we rotate the system to the "interaction basis". In the rotating frame, the time evolution equation (12) reads

$$i\partial_\tau \big|\tilde{\psi}(\tau)\big\rangle = \big(-\tilde{A}_-(t) + \tilde{A}_h(t) + \tilde{A}_r(t)\big)\big|\tilde{\psi}(\tau)\big\rangle, \tag{14}$$

$$\text{where } \big|\tilde{\psi}(\tau)\big\rangle = e^{iG\tau}|\psi(\tau)\rangle, \tag{15}$$

$$\tilde{A}_i(\tau) = e^{iG\tau} A_i e^{-iG\tau}, \quad i = -, h, r. \tag{16}$$

That is, we need to solve a time-dependent Schrödinger equation. The advantage is that now the spectral norm of the time-dependent Hamiltonian, $-\tilde{A}_-(t) + \tilde{A}_h(t) + \tilde{A}_r(t)$, is polylog($N$). Then, we can apply Low and Wiebe's algorithm for time-dependent Hamiltonian simulation,

which is based on truncated Dyson series [25]. The only requirement left to fulfill is to block-encode the time-dependent Hamiltonian. We show how to do this for each of the three terms separately, and then combine them with the Linear Combination of Unitaries technique [21]. Having solved (14), we can invert equation (15) to rotate back to the original basis, effectively retrieving $|\psi(t)\rangle$, and thus efficiently solving the quantum simulation of $A$.

# 3 Preliminaries

To facilitate the reading of our work, in this section we present a summary of well-known concepts and techniques in the literature of quantum algorithms that are essential to understanding our constructions. The proofs of the related theorems can be found in the original papers.

## 3.1 Block-encodings

Block-encodings are arguably the most natural way to handle non-unitary operations with quantum circuits [24, 26]. Given an $n$-qubit matrix A, we consider a $(n + m)$-qubit unitary $U_A$ such that

$$U_A = \frac{1}{\alpha}\begin{bmatrix} A & \cdot \\ \cdot & \cdot \end{bmatrix}, \tag{17}$$

with the $\cdot$ denoting arbitrary matrices. That is, $U_A$ acts as $A$ (up to a normalization factor $\alpha$) controlled on the $m$-qubit ancillary system reading 0,

$$U_A |0^m\rangle |\psi\rangle = |0^m\rangle \left(\frac{A}{\alpha}|\psi\rangle\right) + |\bot\rangle |\psi'\rangle, \tag{18}$$

where $\langle 0^m | \bot \rangle = 0$. More formally,

**Definition 3.1.** Let $A$ be a n-qubit matrix and $\alpha, \epsilon > 0$. A $(m + n)$-qubit unitary $U_A$ is a $(\alpha, m, \epsilon)$-block-encoding of $A$ if

$$\|A - \alpha(\langle 0^m | \otimes I_n) U_A (|0^m\rangle \otimes I_n)\| \le \epsilon. \tag{19}$$

We write $(\alpha, m)$-block-encoding as a shortcut for $(\alpha, m, 0)$-block-encoding.

## 3.2 Linear combination of unitaries

In various situations, it is useful to prepare linear combinations of a set of unitaries $\{U_i\}_{i=0}^{K-1}$,

$$T = \sum_{i=0}^{K-1} y_i U_i. \tag{20}$$

We define two types of operators for this problem:

1. "Select oracle", $U$, such that

$$U := \sum_{i=0}^{K-1} |i\rangle\langle i| \otimes U_i. \tag{21}$$

This operator implements $U_i$ controlled on the $k$ ancilla qubits reading $|i\rangle$.

2. "Prepare oracle", $V$, such that

$$V \left|0^k\right\rangle := \frac{1}{\sqrt{\|\mathbf{y}\|_1}} \sum_{i=0}^{K-1} \sqrt{y_i} \left|i\right\rangle . \tag{22}$$

If the coefficients $y_i$ are not just positive real numbers, then we also need an operator $V'$ such that

$$V'^\dagger \left|0^k\right\rangle := \frac{1}{\sqrt{\|\mathbf{y}\|_1}} \sum_{i=0}^{K-1} \sqrt{y_i}^* \left|i\right\rangle . \tag{23}$$

Note that if the coefficients are real and positive, then $V' = V^\dagger$ suffices.

**Theorem 3.1** (Linear combination of unitaries [21, 27, 28]). *Let* $\mathbf{y} = (y_1, y_2, \ldots, y_K)$ *and let* $\{U_i\}_{i=0}^K$ *be n-qubit unitaries. Then,*

$$W = (V' \otimes I_n) \cdot U \cdot (V \otimes I_n) \tag{24}$$

*is a* $(\|\mathbf{y}\|_1, \lceil \log K \rceil)$*-block encoding of* $T$.

In particular, if $U_i$ are $(\alpha_i, m)$-block-encoding of matrices $A_i$, we can prepare a block encoding of $\sum_{j=0}^k y_j A_j$ using the same method by replacing $y_i$ by $y_i \alpha_i$ in equation (20).

**Corollary 3.1.1.** *Let* $\mathbf{y} = (y_1, y_2, \ldots, y_k)$ *and* $\boldsymbol{\alpha} = (\alpha_1, \alpha_2, \ldots, \alpha_k)$. *Let* $T = \sum_{j=0}^k y_j A_j$, *and let* $U_j$ *be* $(a_j, m, \epsilon)$*-block-encodings of* $A_j$ *for each* $j$. *Then, there is a* $(\|\mathbf{y} \odot \boldsymbol{\alpha}\|_1, m + \lceil \log K \rceil, \|\mathbf{y}\|_1 \epsilon)$*-block-encoding of* $T$ *requiring* 1 *call of the select oracle and one call to each prepare oracle.*

### 3.3 Fast-forwarding

Suppose that we have access to a unitary $U_H$ that acts as an $(\alpha, m)$-block-encoding of an hermitian matrix $H$. The Hamiltonian simulation of $H$ can be viewed as a tranformation of the eigenvalues of $H$ from $\lambda$ to $\exp(-i\lambda t)$, and this mapping can be approximated up to desired precision by a polynomial . With a technique known as qubitization, we can manipulate the eigenvalues to effectively reach any polynomial transformation on $H$. This idea leads to Hamiltonian simulation algorithms that call $U_H$ $\tilde{\mathcal{O}}(\alpha|t|)$ times [22, 24].

An inherent limitation of this approach is that the simulation time of $H$ always scales as $\sim \alpha$, which is lower bounded by the spectral norm of $H$. While qubitization algorithms are provenly optimal for sparse Hamiltonians, they may be inefficient for other classes of Hamiltonians. For example, consider the Hamiltonian $H = cI$, where $I$ is the identity matrix and $c$ is an arbitrary constant. There is a trivial circuit that simulates $H$ with a single gate, despite the spectral norm of $H$ being $c$. Whenever there is an algorithm that prepares $\exp(-iHt)$ faster than $\|Ht\|$, we say that $H$ is *fast-forwardable* [29].

### 3.4 Hamiltonian simulation in the interaction picture

Suppose we want to solve an Hamiltonian simulation problem with $H = H_1 + H_2 \in \mathbb{C}^{N \times N}$, where $\|H_1\| = \Omega(\text{poly} N)$ and $\|H_2\| = \mathcal{O}(\text{polylog} N)$. Clearly, $\|H\| \sim \text{poly} N$, and so we cannot efficiently simulate $H$ with qubitization. But suppose that $H_1$ is fast-forwardable. Then, we can efficiently perform the transformations

$$\left|\psi(\tau)\right\rangle \to \left|\tilde{\psi}(\tau)\right\rangle = e^{iH_1\tau} \left|\psi(\tau)\right\rangle \tag{25}$$

$$\text{and } H_2 \to \tilde{H}_2(\tau) = e^{iH_1\tau} H_2 e^{-iH_1\tau} . \tag{26}$$

The Schrödinger equation in the rotated frame reads

$$i\partial_\tau \left|\tilde{\psi}(\tau)\right\rangle = \tilde{H}_2(\tau)\left|\tilde{\psi}(\tau)\right\rangle. \tag{27}$$

That is, now we need to solve a time-dependent Hamiltonian simulation problem, but with the key advantage that now the spectral norm of the (time-dependent) Hamiltonian is in $\mathcal{O}(\text{polylog}\,N)$.

This problem was solved by Low and Wiebe [25]. The idea starts by expanding the time evolution operator, call it $\mathcal{U}(t)$, into a Dyson series,

$$\mathcal{U}(t) = I - i\int_0^t dt_1\tilde{H}_2(t_1) - \int_0^t dt_2\int_0^{t_2} dt_1\tilde{H}_2(t_2)\tilde{H}_2(t_1) \tag{28}$$
$$+ i\int_0^t dt_3\int_0^{t_3} dt_2\int_0^{t_2} dt_1\tilde{H}_2(t_3)\tilde{H}_2(t_2)\tilde{H}_2(t_1) + \dots$$

Then, we truncate the series to the first $K$ terms and approximate each term by discretizing the time domain into $D$ steps,

$$C_k := \frac{1}{D^k}\sum_{d_k=0}^{D-1}\sum_{d_{k-1}=0}^{d_k}\cdots\sum_{d_1=0}^{d_2}\tilde{H}_2\left(\frac{d_k}{D}t\right)\tilde{H}_2\left(\frac{d_{k-1}}{D}t\right)\dots\tilde{H}_2\left(\frac{d_1}{D}t\right) \tag{29}$$

$$\mathcal{U}(t) \approx \sum_{k=0}^K (-it)^k C_k. \tag{30}$$

Low and Wiebe [25] characterize how $K$ and $D$ need to scale for a precise approximation. Moreover, developing on the ideas of Berry *et al* [21], they show how to construct the terms $C_k$ with linear combinations of operators of the form

$$\sum_{d=0}^{D-1}|d\rangle\langle d|\otimes\tilde{H}_2\left(\frac{d}{D}\tau\right). \tag{31}$$

Their main result is summarized in the Theorem below.

**Theorem 3.2** (Hamiltonian simulation in the interaction picture [25]). *Let* $H_1, H_2 \in \mathbb{C}^{N\times N}$, *and* $\alpha_1$ *and* $\alpha_2$ *be known constants such that* $\|H_1\| \le \alpha_1$ *and* $\|H_2\| \le \alpha_2$. *Let* $\epsilon, t > 0$. *Let* $O_\tau$ *be an* $\left(1, m, \mathcal{O}\left(\frac{\epsilon}{\alpha_2 t}\right)\right)$-*block-encoding of* $\exp(-iH_1\tau)$ *and let* $U_{\tau,D}$ *be an* $\left(\alpha_2, m, \mathcal{O}\left(\frac{\epsilon}{\alpha_2 t}\frac{\log\log(\alpha_2 t/\epsilon)}{\log(\alpha_2 t/\epsilon)}\right)\right)$-*block-encoding of*

$$\sum_{d=0}^{D-1}|d\rangle\langle d|\otimes e^{iH_1\frac{d}{D}\tau}H_2 e^{-iH_1\frac{d}{D}\tau}. \tag{32}$$

*Then, choosing* $\tau = \mathcal{O}(\alpha_2^{-1})$ *and* $D = \mathcal{O}\left(\frac{t}{\epsilon}(\alpha_1 + \alpha_2)\right)$, *we can implement* $\exp(-i(H_1 + H_2)t)$ *up to error* $\epsilon$ *with error probability at most* $\mathcal{O}(\epsilon)$ *with*

1. $\mathcal{O}(\alpha_2 t)$ *calls to* $O_\tau$,

2. $\mathcal{O}\left(\alpha_2 t\frac{\log(\alpha_2 t/\epsilon)}{\log\log(\alpha_2 t/\epsilon)}\right)$ *calls to* $U_{\tau,M}$,

3. $\mathcal{O}\left(\alpha_2 t\left(m + \log\left(\frac{t}{\epsilon}(\alpha_1 + \alpha_2)\right)\right)\log(\alpha_2 t/\epsilon)\right)$ *primitive two-qubit gates*.

## 3.5 Amplitude amplification

Amplitude amplification is a widely used technique in quantum algorithms that generalizes Grover's search [30]. Say we have a quantum algorithm $U$ that prepares a state $a|\psi_G\rangle + \sqrt{1-a^2}|\psi_B\rangle$, and assume that we can quickly distinguish $|\psi_G\rangle$ from $|\psi_B\rangle$. We would like to prepare $|\psi_G\rangle$. We could measure the outcome of $U$ in the $\{|\psi_G\rangle, |\psi_B\rangle\}$ basis until we see $|\psi_G\rangle$. This classical sampling strategy would require calling $U$ an expected number of $\mathcal{O}(a^{-2})$ times. Fortunately, quantum computing allows for a more efficient algorithm.

**Theorem 3.3** (Fixed-point amplitude amplification [24,31]). *Let $|\psi_0\rangle$ be an $n$-qubit state, $U$ be a unitary and $\Pi$ an orthogonal projector acting on $n$ qubits such that*

$$\Pi U|\psi_0\rangle = a|\psi_G\rangle\,, \tag{33}$$

*for some $|\psi_G\rangle$. Then, for any $\epsilon \in\,]0,1]$ and $\delta \in\,]0,a[$, there is a $(n+1)$-qubit unitary $\tilde{U}$ such that*

$$\|\tilde{U}|\psi_0\rangle - |\psi_G\rangle\| \le \epsilon\,, \tag{34}$$

*that consists of $\mathcal{O}(\delta^{-1}\log(1/\epsilon))$ $U$, $U^\dagger$, $C_\Pi \mathrm{NOT}$, $C_{|\psi_0\rangle\langle\psi_0|}\mathrm{NOT}$[3] and two-qubit gates.*

**Corollary 3.3.1.** *Let $U_A$ be an $(\alpha, m)$-block-encoding of some unitary matrix $A$. Then, for any $\epsilon \in\,]0,1]$ and $\delta > \alpha$, we can a build a $(1, m+1, \epsilon)$-block-encoding of $A$ with $\mathcal{O}(\delta\log(1/\epsilon))$ calls to $U$ and $U^\dagger$ and two-qubit gates.*

# 4 Oracle conversions

In our input model, we assumed access to oracles $O_A$, $O_L$, and $O_H$. Below we show how we can build some operators that convey information about the graph that will be useful to our constructions using just $\mathcal{O}(\log(N))$ calls to the input oracles. That is, we could have assumed them to be part of the input model without affecting the final conclusions up to $\mathrm{polylog}(N)$ factors.

**Lemma 4.1.** *There is an operator, call it $O_K$, that acts as*

$$O_K|i,z\rangle = \begin{cases} |i,z\rangle\,, & \text{if } i \text{ is regular node}\,, \\ |i,z\oplus 1\rangle\,, & \text{if } i \text{ is hub}\,, \end{cases} \tag{35}$$

*with $i \in [N]$ and $z \in \{0,1\}$, and can be implemented with $\mathcal{O}(\log N)$ calls to $O_L$.*

*Proof.* According to our input model, if $l$ is larger than the degree of a node $i$, then $r(l,i)$ contains a flag indicating so. So, we can run a binary search, beginning at $N$, to find the largest $l$ such that $r(l,i)$ is not a flag state. This corresponds to the degree of $i$. If $i$ is larger than $N-h$, then $i$ is a hub (*cf.* Definition 2.2). This strategy only requires $\mathcal{O}(\log N)$ calls to $O_L$. $\quad\square$

If we asked $O_D$ to be part of the input model, we could relax the condition that $r(l,i)$ contains a flag if there are less than $l$ non-zero entries in the $i$-th row. Instead, provided with $O_D$, for such $l$'s the function $r(l,i)$ could be take any value without affecting the algorithm.

**Lemma 4.2.** *Let $i$ be a hub. There is an operator, call it $O_Z$, that acts as*

$$O_Z|i,l\rangle = |i,q(l,i)\rangle\,, \tag{36}$$

*with $i,l \in [N]$, where $q(l,i)$ is the position of the $l$-th zero entry in the $i$-th row of $A$, and can be implemented with $\mathcal{O}(\log N)$ calls to $O_L$.*

---

[3]For a projector $\Pi$, we mean $C_\Pi \mathrm{NOT} := \Pi \otimes X + (I - \Pi) \otimes I$.

Table 1: For convenience of the reader, we gather here the operators defined so far. $r(l,i)$ denotes the position of the $l$-th non-zero entry in the $i$-th row of $A$, $q(l,i)$ the position of the $l$-th non-zero entry in the $i$-th row of $A$ when $i$ is a hub, and $h(l)$ the $l$-th hub in the network. $O_A$, $O_L$, and $O_H$ are part of the input model (section 2.2), while $O_K$ and $O_Z$ can be constructed from the input model with a $\log(N)$ overhead (Lemmas 4.1 and 4.2).

| Operator | Description | Expression | Calls to input model |
|---|---|---|---|
| $O_A$ | Accesses adjacency matrix | $O_A\lvert i,j,z\rangle = \lvert i,j,z\oplus A_{ij}\rangle$ | 1 |
| $O_L$ | Accesses adjacency list | $O_L\lvert i,l\rangle = \lvert i,r(l,i)\rangle$ | 1 |
| $O_H$ | Locates hubs | $O_H\lvert l\rangle = \lvert h(l)\rangle$ | 1 |
| $O_K$ | Indicates type of nodes | $O_A\lvert i,j,z\rangle = \lvert i,j,z\oplus A_{ij}\rangle$ | $\mathcal{O}(\log(N))$ |
| $O_Z$ | Locates missing links for hubs | $O_Z\lvert i,l\rangle = \lvert i,q(l,i)\rangle$ | $\mathcal{O}(\log(N))$ |

*Proof.* Let $k_i$ be the degree of node $i$. A fully connected hub $i$ would have $r(l,i) = l$ for all $l \in [N]$, while a hub with missing connections has a total of $N - k_i$ indices missing, leading to discontinuities in the $r(l,i)$ function. The key observation is that we just need to locate these polylog $N$ discontinuities to build the $q(l,i)$ function. We can do this by running a binary search on the array $[r(1,i), r(2,i), \ldots, r(k_i,i)]$, whose entries can be quickly accessed via queries to $O_L$. We know that the positions $r(l,i)+1, r(l,i)+2, \ldots, r(l,i)+x-1$ are zeros in the $i$-th row of $A$ if $r(l+1,i) - r(l,i) = x > 1$.

Each binary search requires $\mathcal{O}(\log N)$ calls to $O_L$. For a hub-sparse network, there are polylog $N$ hubs, each with polylog $N$ zero entries. As such, we can reconstruct the function $q(l,i)$ with polylog $N$ calls to $O_L$. Having classical access to $q(l,i)$, the unitary $O_Z$ can be built with a polylog $N$ overhead. $\qquad\square$

Our simulation algorithm will make use of operators $O_A$, $O_L$, $O_H$, $O_K$, and $O_Z$. We summarize their properties in Table 1.

## 5 Fast-forwarding hubs

As we have discussed in the Main Results section, our strategy is to move the system to the interaction picture. For that, we need to efficiently implement the operation $\exp(-iGt)$, with $G$ being defined as in equation (9). Here we show that this is indeed possible. Note that since $\|G\| = \tilde{\Theta}(\sqrt{N})$, this proves that $G$ is fast-forwardable.

The key is to analyze the spectral decomposition of $G$. The corresponding graph contains $M$ hubs, each connected to all the $N - M$ regular nodes (but with no connections between hubs). Considering this simple structure, we can prove the following.

**Lemma 5.1.** *Let $G$ be defined as in equation (9). Say that there are $M$ hubs and $N - M$ regular nodes. Then, there are only two eigenvectors with eigenvalues different from zero. These are*

$$\lvert\Psi_\pm\rangle = \frac{1}{\sqrt{2}}\sum_{j\text{ hub}}\frac{\lvert j\rangle}{\sqrt{M}} \pm \frac{1}{\sqrt{2}}\sum_{j\text{ regular node}}\frac{\lvert j\rangle}{\sqrt{N-M}}, \tag{37}$$

*and have eigenvalues*

$$\lambda_\pm = \sqrt{M(N-M)}. \tag{38}$$

*Proof.* That $|\Psi_\pm\rangle$ are eigenvectors with eigenvalues $\lambda_\pm$ can be proved by direct inspection. To see that all the other eigenvalues are zero, we show in Appendix A that the characteristic polynomial of $G$ is (up to a sign)

$$\lambda^{N-2}\left(\lambda^2 - M(N-M)\right). \tag{39}$$

Therefore, besides $\lambda_\pm$ the only acceptable solutions are $\lambda = 0$. □

Consequently, we can write

$$e^{-iGt} = (e^{-i\lambda_+ t} - 1)|\Psi_+\rangle\langle\Psi_+| + (e^{-i\lambda_- t} - 1)|\Psi_-\rangle\langle\Psi_-| + I. \tag{40}$$

So, we can simulate $G$ by preparing $|\Psi_\pm\rangle$ and applying the corresponding phase $(-\lambda_\pm t)$. First, we show that the states can be prepared efficiently.

**Lemma 5.2.** *Let $|\Psi_\pm\rangle$ be defined as in Lemma 5.1, and let $P_\pm$ be operators acting as*

$$P_\pm |0^n\rangle = |\Psi_\pm\rangle. \tag{41}$$

*Then, we can prepare* $(\text{polylog}\,N, 2)$*-block-encodings of $P_\pm$ with $\mathcal{O}(1)$ calls to the controlled versions of $O_K$ and $O_H$ and $\mathcal{O}(\text{polylog}\,N)$ primitive two-qubit gates.*

*Proof.* The states $|\Psi_\pm\rangle$ are combinations of a uniform superposition over the hub nodes and a uniform superposition over the regular nodes. To prepare a uniform superposition over the regular nodes, we do

$$|0\rangle|0^n\rangle \xrightarrow{H^{\otimes n}} |0\rangle \sum_{j=0}^{N-1} \frac{|j\rangle}{\sqrt{N}} \tag{42}$$

$$\xrightarrow{O_K} |0\rangle \sum_{j \text{ regular node}} \frac{|j\rangle}{\sqrt{N}} + |1\rangle \sum_{j \text{ hub}} \frac{|j\rangle}{\sqrt{N}} \tag{43}$$

$$= \sqrt{\frac{N-M}{N}} |0\rangle \sum_{j \text{ regular node}} \frac{|j\rangle}{\sqrt{N-M}} + |\bot\rangle, \tag{44}$$

where $\langle 0|\bot\rangle = 0$. That is, $O_K H^{\otimes n}$ is a $\left(\sqrt{\frac{N}{N-M}}, \log M\right)$-block-encoding of a preparation of the uniform superposition of regular nodes. A uniform superposition of hubs can be prepared as

$$|0^n\rangle \xrightarrow{H^{\otimes \log M}} \sum_{j=0}^{M-1} \frac{|j\rangle}{\sqrt{M}} \tag{45}$$

$$\xrightarrow{O_H} \sum_{j=0}^{M-1} \frac{|h(j)\rangle}{\sqrt{M}}. \tag{46}$$

Then, $P_\pm$ can be expressed as

$$\frac{1}{\sqrt{2}} \sqrt{\frac{N}{N-M}} O_K H^{\otimes n} \pm \frac{1}{\sqrt{2}} O_H H^{\otimes \log M}. \tag{47}$$

From Theorem 3.1, we can do this with the Linear Combination of Unitaries technique and we get a $\left(\frac{1}{\sqrt{2}} + \frac{1}{\sqrt{2}}\sqrt{\frac{N}{N-M}}, 2\right)$-block-encoding of $P_\pm$. □

Now it becomes straightforward how to simulate $\exp(-iGt)$.

**Lemma 5.3.** *Let $G$ be defined as in equation (9). Then, we can prepare a $(1, 8, \epsilon)$-block-encoding of $\exp(-iGt)$ with $\mathcal{O}(\mathrm{polylog}(N)\log(1/\epsilon))$ calls to the controlled and inverse versions of $O_K$ and $O_H$ and primitive two-qubit gates.*

*Proof.* Consider the following circuit. The "$a$", "$-$", and "$+$" are labels of the ancilla registers. $U_{\pm}$ are $(\mathrm{polylog}\, N, 2)$-block-encoding of $P_{\pm}$, as per Lemma 5.2, and the corresponding ancilla qubits are $\pm$.

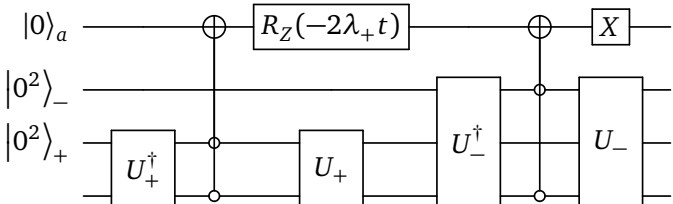

In Appendix A we show by direct inspection that this circuit acts as a $(\beta, 5)$-bock-encoding of

$$e^{-i\lambda_+ t} |\Psi_+\rangle\langle\Psi_+| + e^{-i\lambda_- t} |\Psi_-\rangle\langle\Psi_-| \,, \tag{48}$$

for

$$\beta = \frac{1}{2}\left(1 + \sqrt{\frac{N}{N-M}}\right)^2 \,. \tag{49}$$

For the same reason, the circuit

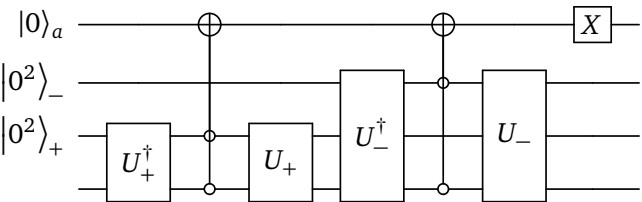

is a $(\beta, 5)$-bock-encoding of

$$|\Psi_+\rangle\langle\Psi_+| + |\Psi_-\rangle\langle\Psi_-| \,. \tag{50}$$

Finally, the empty circuit is a $(1, 5)$-bock-encoding of the identity, $I$.

Let us denote these three circuits as $U_1$, $U_2$, and $U_I$. By equation (40), we want to implement

$$\beta U_1 - \beta U_2 + U_I \,, \tag{51}$$

which we can do with Linear Combination of Unitaries. From Lemma 3.1, we get a $(2\beta + 1, 7)$-block-encoding of $\exp(-iGt)$ with $\mathcal{O}(1)$ calls to the controlled and inverse versions of $O_K$ and $O_H$ plus $\mathcal{O}(\mathrm{polylog}(N))$ two-qubit gates.

Finally, we can amplify the probability of measuring the right block with amplitude amplification. From Corollary 3.3.1, we can prepare a $(1, 8, \epsilon)$-block-encoding of $\exp(-iGt)$ with $\mathcal{O}(\beta \log(1/\epsilon))$ calls to the stated gates. □

## 6 Hub-sparse networks in the interaction picture

We have seen that we can efficiently implement $\exp(-iGt)$. Therefore, we can quickly rotate to the interaction picture – equation (14). We are left with the task of solving the Schrödinger equation with the time-dependent Hamiltonian $e^{iGt}(-A_- + A_h + A_r)e^{-iGt}$.

As we have discussed in the [Preliminaries] section, Low and Wiebe have solved this problem by approximating the evolution operator with a truncated Dyson series [25]. According to Theorem 3.2, all that is left for us to do is to ensure that there is an efficient block-encoding of

$$\sum_{d=0}^{D-1} |d\rangle\langle d| \otimes \left( e^{iG\frac{d}{D}\tau} A_i e^{-iG\frac{d}{D}\tau} \right), \quad \text{for } i = -, h, r. \tag{52}$$

We show that this is indeed the case in two steps. First, for $i = -, h, r$ we show that $A_i$ is efficiently block-encodable. Then, we show that we can build a block-encoding of the operator (52) from a block-encoding of $A_i$.

**Lemma 6.1.** *We can prepare* $(\text{polylog}\,N, n+4)$*-block-encodings of $A_-$, $A_h$, and $A_r$ with $\mathcal{O}(1)$ calls to $O_A$, $O_L$, $O_H$, $O_Z$, $O_K$, and $\mathcal{O}(\text{polylog}\,N)$ primitive two-qubit gates.*

*Proof.* In Appendix B we explicitly describe polylog($N$)-sized circuits that block-encode $A_-$, $A_h$, and $A_r$. The idea is to adapt a standard construction of block-encodings sparse matrices ( [32, Chapter 6], for example) to our situation. For $A_h$ and $A_r$ we include a step that checks if we are linking hubs to hubs and regular nodes to regular nodes, respectively. For $A_-$ we use the operator $O_Z$ to switch the roles of the "0"s and "1"s in the usual technique for block-encoding sparse matrices. For that case, we also add a routine that imposes that we are transitioning from a hub to a regular node or vice-versa. □

**Lemma 6.2.** *Let $U_i$ be a $(\alpha_i, m)$-block-encoding of $A_i$. Then, we can prepare a $(\alpha_i, m+16, \epsilon)$-block-encoding of the operator in equation (52) with*

$$\mathcal{O}(\log(D)\,\text{polylog}(N)\log(1/\epsilon)) \tag{53}$$

*calls to $U_i$, to the controlled and inversed versions of $O_H$ and $O_K$, and to primitive two-qubit gates.*

*Proof.* We adapt a construction from [22, Theorem 7]. Let $\mathcal{G}_t$ be a $\left(1, 8, \frac{\epsilon}{2\log D}\right)$-block-encoding of $\exp(-iGt)$ – from Theorem 5.3 we can prepare this with $\mathcal{O}(\text{polylog}(N)\log(1/\epsilon))$ calls to the controlled and inverse versions of $O_K$ and $O_H$ and to two-qubit gates (independently of $t$). Now let $C\mathcal{G}$ be the unitary implemented by the following circuit.

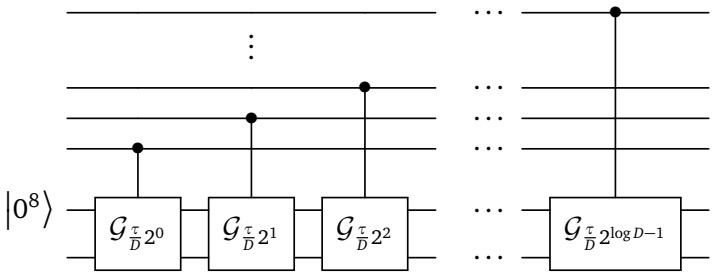

If the first $\log D$ qubits are in state $|d\rangle \equiv |d_{\log D-1} \ldots d_2 d_1 d_0\rangle$, then the last $n$ qubits are transformed by

$$\mathcal{G}_{\frac{\tau}{D}2^0 d_0} \mathcal{G}_{\frac{\tau}{D}2^1 d_1} \mathcal{G}_{\frac{\tau}{D}2^2 d_2} \mathcal{G}_{\frac{\tau}{D}2^{\log D-1} d_{\log D-1}}. \tag{54}$$

Since $\mathcal{G}_t$ differs from $\exp(-iGt)$ by at most $\epsilon/(2\log D)$, we can guarantee (by a telescopic triangle inequality) that

$$\left\| \mathcal{G}_{\frac{\tau}{D}2^0 d_0} \mathcal{G}_{\frac{\tau}{D}2^1 d_1} \mathcal{G}_{\frac{\tau}{D}2^2 d_2} \mathcal{G}_{\frac{\tau}{D}2^{\log D-1} d_{\log D-1}} - e^{-iG\frac{d}{D}\tau} \right\| \leq \frac{\epsilon}{2} \tag{55}$$

$$\Leftrightarrow \left\| \langle 0^8 | C\mathcal{G} | 0^8 \rangle - \sum_{d=0}^{D-1} |d\rangle\langle d| \otimes e^{-iG\frac{d}{D}\tau} \right\| \leq \frac{\epsilon}{2}. \tag{56}$$

Now we want to get the product of the matrices encoded by $C\mathcal{G}^\dagger$, $U_i$, and $C\mathcal{G}$,

$$\left\| \langle 0^8|C\mathcal{G}^\dagger|0^8\rangle \, \langle 0^m|U_i|0^m\rangle \, \langle 0^8|C\mathcal{G}|0^8\rangle - \sum_{d=0}^{D-1}|d\rangle\langle d| \otimes \frac{e^{iG\frac{d}{D}\tau}A_i e^{-iG\frac{d}{D}\tau}}{\alpha_i} \right\| \leq \epsilon . \tag{57}$$

For that purpose, we just need to treat the ancilla qubits separately (as in [24, Lemma 53]), ending up with an $(\alpha_i, 8 + m + 8, \epsilon)$-block-encoding. $\qquad\square$

We finally have all the necessary ingredients to conclude the proof of Theorem 2.1.

*Proof of Theorem 2.1.* We apply Low and Wiebe's [25] result on Hamiltonian simulation in the interaction picture to our problem, making use of the results we have proved so far. We wish to set $H_1 = G$ and $H_2 = -A_- + A_h + A_r$ in Lemma 3.2. In this case, we have that $\alpha_1 = \mathcal{O}(\sqrt{N}\operatorname{polylog}N)$ (Lemma 5.1) and $\alpha_2 = \mathcal{O}(\operatorname{polylog}N)$ ($H_2$ is the sum of three $\operatorname{polylog}(N)$-sparse matrices).

From Lemma 5.3, we can prepare a $\left(1, m, \mathcal{O}\left(\frac{\epsilon}{\alpha_2 t}\right)\right)$-block-encoding of $\exp(-iG\tau)$ in time $\mathcal{O}(\log(\alpha_2 t/\epsilon)\operatorname{polylog}(N))$. By Theorem 3.2, this operator is called $\mathcal{O}(\alpha_2 t)$ times, yielding a total complexity of

$$\mathcal{O}\left(\alpha_2 t \times \log(\alpha_2 t/\epsilon)\operatorname{polylog}N\right) = \mathcal{O}\left(t\,\log(t/\epsilon)\,\operatorname{polylog}(N)\right). \tag{58}$$

From Lemmas 6.1 and 6.2, we can prepare $(\operatorname{polylog}N, n+20, \epsilon')$ block-encodings the operators in equation (52) with $\mathcal{O}(\log(D)\log(1/\epsilon')\operatorname{polylog}(N))$ gates. We can then use Linear Combination of Unitaries (Corollary 3.1.1) to prepare a $(\operatorname{polylog}N, n+22, 3\epsilon')$-block-encoding of

$$\sum_{d=0}^{D-1}|d\rangle\langle d| \otimes \left(e^{iG\frac{d}{D}\tau}(-A_- + A_h + A_r)e^{-iG\frac{d}{D}\tau}\right), \tag{59}$$

with essentially the same resources. This is the $U_{\tau,D}$ operator from Theorem 3.2. We need to call this operator $\mathcal{O}\left(\alpha_2 t \frac{\log(\alpha_2 t/\epsilon)}{\log\log(\alpha_2 t/\epsilon)}\right)$ times. The total complexity from calls to $U_{\tau,D}$ is then

$$\mathcal{O}\left(\alpha_2 t \frac{\log(\alpha_2 t/\epsilon)}{\log\log(\alpha_2 t/\epsilon)} \times \log(D)\log(1/\epsilon')\operatorname{polylog}(N)\right) = \mathcal{O}\left(t\,\frac{\log^3(t/\epsilon)}{\log\log(t/\epsilon)}\operatorname{polylog}(N)\right), \tag{60}$$

where we have set $\epsilon' = \mathcal{O}\left(\frac{\epsilon}{\alpha_2 t}\frac{\log\log(\alpha_2 t/\epsilon)}{\log(\alpha_2 t/\epsilon)}\right)$ and $D = \mathcal{O}\left(\frac{t}{\epsilon}(\alpha_1 + \alpha_2)\right)$, as per Theorem 3.2. $\qquad\square$

# 7 Discussion

In this work, we have introduced hub-sparse networks (Definition 2.2). These are a generalization of sparse networks, allowing for a polylogarithmic number of hubs. We have shown that such networks are also efficiently simulatable (in the sense of Definition 2.1), thus extending the previous literature of quantum algorithms. We believe that this constitutes an important step in applying quantum computing to the study of complex networks.

A point worth discussing is the input model. We have assumed access to an oracle to the adjacency matrix, $O_A$, an oracle to the adjacency list, $O_L$, and an oracle to the location of the hubs, $O_H$. The final result was expressed in terms of calls to these oracles, as well as primitive two-qubit gates. A natural question is what would be the cost of actually building each of these oracles. For "artificial" networks, such as physical Hamiltonians, we have an analytical description of the graph. This means that we have closed expressions for $A_{ij}$, $r(l, i)$,

and $h(l)$ (*cf.* Section 2.2). We can build circuits that evaluate these functions with a size that is independent of the size of the network. In contrast, networks based on real world data need to be pre-processed into a suitable data structure. A common representation in network science is the adjacency (or edge) list model, with the nodes ordered by degree [33]. If we can quickly access the entries of the adjacency list of any given node, then we can also quickly determine $A_{ij}$, $r(l, i)$, and $h(l)$. To get the oracles $O_A$, $O_L$, and $O_H$ we just require that this access is coherent, that is, that we can access the adjacency list data in quantum superposition. Such could be achieved with a quantum random access memory (qRAM), a device that would load classical data in coherent superposition in time logarithmic in the number of memory cells [34]. We should nevertheless point out that, even though there have been proposals of physical architectures for implementing qRAM [35, 36], there are still significant challenges to overcome before such a device can be practically realized [37].

A possible caveat to the feasiblity of our input model is that the lists of neighbours of the hub nodes have lengths close to $N$. Therefore, storing them into memory would take $\mathcal{O}(N)$ time per hub, even if we could later access their entries quickly. Fortunately, our algorithm does necessarily require an oracle acting as $O_L$ at the hub nodes. Instead, for these nodes we just need to identify the non-connections. Therefore, we could assume an input model that, besides access to $O_A$, $O_H$, and $O_K$, demands an oracle $O_L$ that acts as (6) when $i$ is a regular node, an oracle $O_Z$ that acts as (36) when $i$ is a hub. This modifications would imply no change to our algorithm, and have the advantage that all the underlying adjacency lists have polylog($N$) size.

Despite our progress, our model remains an oversimplification of real world complex networks. Our proof was critically dependent on the assumption that each node either had polylog($N$) connections or polylog($N$) missing connections. Although the presence of hubs is a recurrent organizational principle of complex networks, we would not expect to find a hub-sparse structure in many real world networks. Instead, we expect a continuous transition between the degree of the least connected nodes to the largest hubs in the network, and it remains open the question of whether such networks can be efficiently simulated on a quantum computer. Standard models for the degree distribution of complex networks include the scale-free model or the log-normal model. Scale-free networks are networks where the degrees of the most connected nodes tend to follow a power-law distribution [2, 38], and this model has dominated the network science literature for almost 20 years. In a recent paper it was proposed that a log-normal distribution might be a better fit for the degree distribution of most real-world complex networks [36]. However, simple variations of the original scale-free model are also a good fit [36]. Previous results have shown that it is not possible to simulate a general $N \times N$ Hamiltonian in polylog $N$ time [39], implying that efficient simulation algorithms must exploit strong structural properties in the system. As such, a natural next step for complex network simulation, and a considerably more challenging problem, would be to consider a network simulation algorithm directly exploiting a power-law or log-normal degree distribution. Developing an algorithm capable of efficiently simulating such a network, or proving that it does not exist, is an open and important question to be addressed in future work.

## Acknowledgments

We would like to thank Leonardo Novo and Yasser Omar for useful suggestions and discussions.

**Funding information** The authors thank Fundação para a Ciência e a Tecnologia (Portugal) for its support through the project UIDB/EEA/50008/2020. JPM and DM acknowledge the support of Fundação para a Ciência e a Tecnologia (FCT, Portugal) through scholarships 2019.144151.BD and 2020.04677.BD, respectively. BC acknowledges the support of FCT through project CEECINST/00117/2018[0]/CP1495/CT0001.

# A  Spectral decomposition of the $G$ matrix

*Proof of Lemma 5.1.* Let $D_N(\lambda)$ denote the characteristic polynomial of the matrix $G$, assuming that there are $N$ nodes and $M$ hubs. Recall that, up to a sign, the determinant is invariant under the exchange of columns/rows. Then, we can write

$$
D_N(\lambda) =
\overbrace{\begin{matrix} & & M & & \end{matrix}}
\left|\begin{array}{ccccc|ccccc}
-\lambda & 0 & 0 & \dots & 0 & 1 & 1 & 1 & \dots & 1 \\
0 & -\lambda & 0 & \dots & 0 & 1 & 1 & 1 & \dots & 1 \\
0 & 0 & -\lambda & \dots & 0 & 1 & 1 & 1 & \dots & 1 \\
\vdots & \vdots & \vdots & \ddots & \vdots & \vdots & \vdots & \vdots & \ddots & \vdots \\
0 & 0 & 0 & \dots & -\lambda & 1 & 1 & 1 & \dots & 1 \\
1 & 1 & 1 & \dots & 1 & -\lambda & 0 & 0 & \dots & 0 \\
1 & 1 & 1 & \dots & 1 & 0 & -\lambda & 0 & \dots & 0 \\
1 & 1 & 1 & \dots & 1 & 0 & 0 & -\lambda & \dots & 0 \\
\vdots & \vdots & \vdots & \ddots & \vdots & \vdots & \vdots & \vdots & \ddots & 0 \\
1 & 1 & 1 & \dots & 1 & 0 & 0 & 0 & \dots & -\lambda
\end{array}\right|
\begin{matrix} \Big\} M \\ \\ \Big\} N-M \end{matrix}
\tag{A.1}
$$

Performing a Laplace expansion on the $(M+1)$-th column, we find

$$
D_N(\lambda) = (-1)^M M F_{N-1}(\lambda) - \lambda D_{N-1}(\lambda), \tag{A.2}
$$

where

$$
F_{N-1}(\lambda) =
\left|\begin{array}{ccccc|cccc}
0 & -\lambda & 0 & \dots & 0 & 1 & 1 & \dots & 1 \\
0 & 0 & -\lambda & \dots & 0 & 1 & 1 & \dots & 1 \\
\vdots & \vdots & \vdots & \ddots & \vdots & \vdots & \vdots & \ddots & \vdots \\
0 & 0 & 0 & \dots & -\lambda & 1 & 1 & \dots & 1 \\
1 & 1 & 1 & \dots & 1 & 0 & 0 & \dots & 0 \\
1 & 1 & 1 & \dots & 1 & -\lambda & 0 & \dots & 0 \\
1 & 1 & 1 & \dots & 1 & 0 & -\lambda & \dots & 0 \\
\vdots & \vdots & \vdots & \ddots & \vdots & \vdots & \vdots & \ddots & 0 \\
1 & 1 & 1 & \dots & 1 & 0 & 0 & \dots & -\lambda
\end{array}\right|
\begin{matrix} \Big\} M-1 \\ \\ \Big\} N-M \end{matrix}
\tag{A.3}
$$

Expanding along the $(M+1)$-th column of the matrix in (A.3), we conclude that

$$
F_{N-1}(\lambda) = -\lambda F_{N-2}(\lambda) \tag{A.4}
$$

$$
= (-\lambda)^{N-1-M} \times
\left|\begin{array}{ccccc}
0 & -\lambda & 0 & \dots & 0 \\
0 & 0 & -\lambda & \dots & 0 \\
\vdots & \vdots & \vdots & \ddots & \vdots \\
0 & 0 & 0 & \dots & -\lambda \\
1 & 1 & 1 & \dots & 1
\end{array}\right|
\Big\} M
\tag{A.5}
$$

$$
= (-\lambda)^{N-1-M} \times \lambda^{M-1}, \tag{A.6}
$$

where we have applied equation (A.4) recursively $N-1-M$ times to reach line (A.5) and we have expanded the determinant along the first column of the matrix of line (A.5) to get to line (A.6).

Combining this into equation (A.2), we get

$$D_N(\lambda) = (-1)^{N-1} M \lambda^{N-2} - \lambda D_{N-1}(\lambda) \tag{A.7}$$

$$= (-1)^{N-1} M(N-M) \lambda^{N-2} + (-\lambda)^{N-M} \left. \begin{vmatrix} -\lambda & 0 & 0 & \dots & 0 \\ 0 & -\lambda & 0 & \dots & 0 \\ 0 & 0 & -\lambda & \dots & 0 \\ \vdots & \vdots & \vdots & \ddots & \vdots \\ 0 & 0 & 0 & \dots & -\lambda \end{vmatrix} \right\} M \tag{A.8}$$

$$= (-1)^{N-1} M(N-M) \lambda^{N-2} + (-\lambda)^{N-M} (-\lambda)^M \tag{A.9}$$

$$= (-1)^N \lambda^{N-2} \left( -M(N-M) + \lambda^2 \right), \tag{A.10}$$

where we have applied equation (A.7) recursively $N-M$ times to move to line (A.8). $\qquad\square$

*Proof of Lemma 5.3*. Let $U$ be the unitary describing the action of the entire circuit. We analyse $\langle 0^5 | U | 0^5 \rangle$ in the diagonal basis and show that it behaves as expected.

- $\langle \Psi_\pm | \langle 0^5 | U | 0^5 \rangle | \Psi_\pm \rangle$.

  Let $\alpha = \text{polylog}\, N$ be the block-encoding factor of $U_\pm$. First, note that

$$U_+ \left| 0^2 \right\rangle_+ |0^n\rangle =: \alpha^{-1} \left| 0^2 \right\rangle_+ |\Psi_+\rangle + \sqrt{1-\alpha^{-2}} |\perp_+\rangle , \tag{A.11}$$

from some $|\perp_+\rangle$ such that $\langle \perp_+ | 0^2 \rangle_+ = 0$ and $\langle \perp_+ | \perp_+ \rangle = 1$. Also,

$$U_+^\dagger |\perp_+\rangle =: \sqrt{1-\alpha^{-2}} \left| 0^2 \right\rangle_+ |0^n\rangle - \alpha^{-1} \left| \perp_+' \right\rangle , \tag{A.12}$$

for some $\left| \perp_+' \right\rangle$ such that $\langle \perp_+' | 0^2 \rangle_+ = 0$ and $\langle \perp_+' | \perp_+' \rangle = 1$. Then, writing $\theta = -\lambda_+ t$,

$$\left| 0^2 \right\rangle_+ |\Psi_+\rangle |0\rangle_a \xrightarrow{U_+^\dagger} \alpha \left( \left| 0^2 \right\rangle_+ |0^n\rangle - \sqrt{1-\alpha^{-2}} U_+^\dagger |\perp_+\rangle \right) |0\rangle_a \tag{A.13}$$

$$= \alpha^{-1} \left| 0^2 \right\rangle_+ |0^n\rangle |0\rangle_a + \sqrt{1-\alpha^{-2}} \left| \perp_+' \right\rangle |0\rangle_a \tag{A.14}$$

$$\xrightarrow{\text{CCX} \cdot R_Z(2\theta)} e^{i2\theta} \alpha^{-1} \left| 0^2 \right\rangle_+ |0^n\rangle |1\rangle_a + e^{-i2\theta} \sqrt{1-\alpha^{-2}} \left| \perp_+' \right\rangle |0\rangle_a \tag{A.15}$$

$$\xrightarrow{U_+} e^{i\theta} \alpha^{-1} \left( \alpha^{-1} \left| 0^2 \right\rangle_+ |\Psi_+\rangle + \sqrt{1-\alpha^{-2}} |\perp_+\rangle \right) |1\rangle_a \tag{A.16}$$

$$+ e^{-i\theta} \sqrt{1-\alpha^{-2}} U_+ \left| \perp_+' \right\rangle |0\rangle_a \tag{A.17}$$

$$= e^{i\theta} \alpha^{-1} \left( \alpha^{-1} \left| 0^2 \right\rangle_+ |\Psi_+\rangle + \sqrt{1-\alpha^{-2}} |\perp_+\rangle \right) |1\rangle_a \tag{A.18}$$

$$+ e^{-i\theta} \sqrt{1-\alpha^{-2}} \left( \sqrt{1-\alpha^{-2}} \left| 0^2 \right\rangle_+ |\Psi_+\rangle - \alpha^{-1} |\perp_+\rangle \right) |0\rangle_a . \tag{A.19}$$

The second part of the circuit is not going to affect the $+$ register. So, only the part of the state with $\left| 0^2 \right\rangle_+$ at this point will contribute to the block encoding, that is,

$$\left| 0^2 \right\rangle_+ |\Psi_+\rangle \left( e^{i\theta} \alpha^{-2} |1\rangle_a + e^{-i\theta}(1-\alpha^{-2}) |0\rangle_a \right) . \tag{A.20}$$

Now notice that

$$\left\langle 0^2 \right|_- \langle 0^n | U_-^\dagger \left| 0^2 \right\rangle_- |\Psi_+\rangle = 0 , \tag{A.21}$$

because $\langle \Psi_- | \Psi_+ \rangle = 0$. Therefore, the second CCX operator does not affect this part of the state and the $U_-$ operator will just undo the effect of $U_-^\dagger$. We conclude that

$$\langle 0|_a \langle 0|_- \langle 0|_+ \langle \Psi_+ | U |0\rangle_+ |\Psi_+\rangle |0\rangle_- |0\rangle_a = e^{i\theta} \alpha^{-2} . \tag{A.22}$$

The case for $|\Psi_-\rangle$ is similar, but the $a$ register reaches the $R_Z$ gate in state $|0\rangle$, and so we get a $-\theta$ phase.

- $\langle\phi|\langle 0^5|U|0^5\rangle|\phi_0\rangle$, for any eigenvectors $\phi,\phi_0$ such that $G|\phi_0\rangle = 0$.
  We start by noting that

$$\langle 0^n|\langle 0^2|_+ U_+^\dagger |0^2\rangle|\phi_0\rangle = \left(\alpha^{-1}\langle 0^2|_+ \langle\Psi_+| + \sqrt{1-\alpha^{-2}}\langle\perp_+|\right)|0^2\rangle|\phi_0\rangle = 0. \quad (A.23)$$

So, the controlled-controlled-X gate has no effect and the $U_+$ operator just undoes the action of $U_+^\dagger$. The same reasoning applies to the second part of the circuit. We conclude that the $a$ register ends up in state $|1\rangle_a$ with probability 1, and so $\langle 0|_a U|0^5\rangle|\phi_0\rangle = 0$. In particular, it follows that $\langle\phi|\langle 0^5|U|0^5\rangle|\phi_0\rangle = 0$.

$\square$

# B  Block-encoding the matrices $A_-$, $A_h$, and $A_r$

*Proof of Lemma 6.1.* For each of these matrices, we write down a block-encoding circuit and show that it acts as desired.

- Block-encoding circuit for $A_h$:

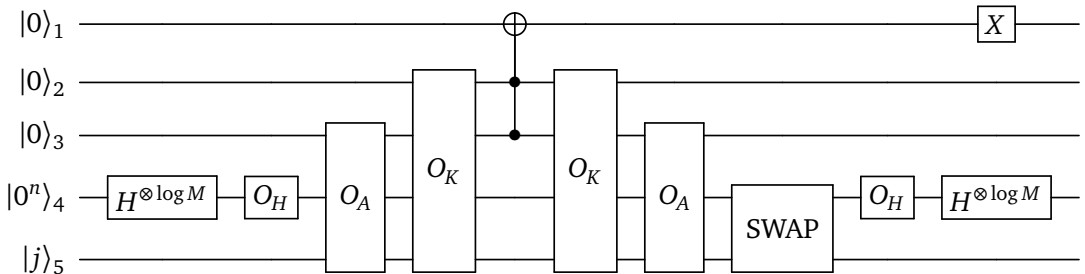

where register "5" has $n$ qubits and $O_K$ is acting on the registers "2" and "5".

If $U$ is the unitary implemented by this circuit, we want to show that

$$\langle 0^{n+3}|_{1,2,3,4}\langle i|_5 U|0^{n+3}\rangle_{1,2,3,4}|j\rangle_5 \propto A_h|_{i,j}. \quad (B.1)$$

First, observe that

$$|0\rangle_1 |0\rangle_2 |0\rangle_3 |0^n\rangle_4 |j\rangle_5 \xrightarrow{H^{\otimes\log M}} |0\rangle_1 |0\rangle_2 |0\rangle_3 \sum_{l=0}^{M-1}\frac{|l\rangle_4}{\sqrt{M}}|j\rangle_5 \quad (B.2)$$

$$\xrightarrow{O_H} |0\rangle_1 |0\rangle_2 |0\rangle_3 \sum_{l=0}^{M-1}\frac{|h(l)\rangle_4}{\sqrt{M}}|j\rangle_5 \quad (B.3)$$

$$\xrightarrow{O_A} |0\rangle_1 |0\rangle_2 \sum_{l=0}^{M-1}\left|A_{h(l),j}\right\rangle_3 \frac{|h(l)\rangle_4}{\sqrt{M}}|j\rangle_5 \quad (B.4)$$

$$\xrightarrow{O_K} |0\rangle_1 |j\text{ hub?}\rangle_2 \sum_{l=0}^{M-1}\left|A_{h(l),j}\right\rangle_3 \frac{|h(l)\rangle_4}{\sqrt{M}}|j\rangle_5 \quad (B.5)$$

$$\xrightarrow{\text{Toffoli}} \sum_{l=0}^{M-1}\left|A_h|_{h(l),j}\right\rangle_1 |j\text{ hub?}\rangle_2 \left|A_{h(l),j}\right\rangle_3 \frac{|h(l)\rangle_4}{\sqrt{M}}|j\rangle_5 \quad (B.6)$$

$$\xrightarrow{O_K} \sum_{l=0}^{M-1}\left|A_h|_{h(l),j}\right\rangle_1 |0\rangle_2 \left|A_{h(l),j}\right\rangle_3 \frac{|h(l)\rangle_4}{\sqrt{M}}|j\rangle_5 \quad (B.7)$$

$$\xrightarrow{O_A} \sum_{l=0}^{M-1}\left|A_h|_{h(l),j}\right\rangle_1 |0\rangle_2 |0\rangle_3 \frac{|h(l)\rangle_4}{\sqrt{M}}|j\rangle_5. \quad (B.8)$$

Since we are only interested in the final state when all ancilla qubits are the zero state, we may consider the action of the final three gates on $\left|0^{n+3}\right\rangle_{1,2,3,4}|i\rangle_5$,

$$|0\rangle_1 |0\rangle_2 |0\rangle_3 |0^n\rangle_4 |i\rangle_5 \xrightarrow{X} |1\rangle_1 |0\rangle_2 |0\rangle_3 |0^n\rangle_4 |i\rangle_5 \tag{B.9}$$

$$\xrightarrow{H^{\otimes \log M}} |1\rangle_1 |0\rangle_2 |0\rangle_3 \sum_{l'=0}^{M-1} \frac{|l'\rangle_4}{\sqrt{M}} |i\rangle_5 \tag{B.10}$$

$$\xrightarrow{O_H} |1\rangle_1 |0\rangle_2 |0\rangle_3 \sum_{l'=0}^{M-1} \frac{|h(l')\rangle_4}{\sqrt{M}} |i\rangle_5 \tag{B.11}$$

$$\xrightarrow{\text{SWAP}} |1\rangle_1 |0\rangle_2 |0\rangle_3 \sum_{l'=0}^{M-1} |i\rangle_4 \frac{|h(l')\rangle_5}{\sqrt{M}}. \tag{B.12}$$

Then, the inner product yields (where $U$ is the unitary implemented by this circuit)

$$\left\langle 0^{n+3}\right|_{1,2,3,4} \langle i|_5 U \left|0^{n+3}\right\rangle_{1,2,3,4} |j\rangle_5 = \frac{1}{M} \sum_{l,l'} A_h|_{h(l),j} \delta_{i,h(l)} \delta_{j,h(l')} \tag{B.13}$$

$$= \frac{1}{M} A_h|_{i,j}. \tag{B.14}$$

We have used that, if $A_h|_{i,j} \neq 0$, there is a unique $l$ such that $h(l) = i$ and that, in that case, there is also a unique $l'$ such that $h(l') = j$. If $A_h|_{i,j} = 0$, there are no such $l$ and $l'$, and so (B.13) sums to zero.

- Block-encoding circuit for $A_r$:

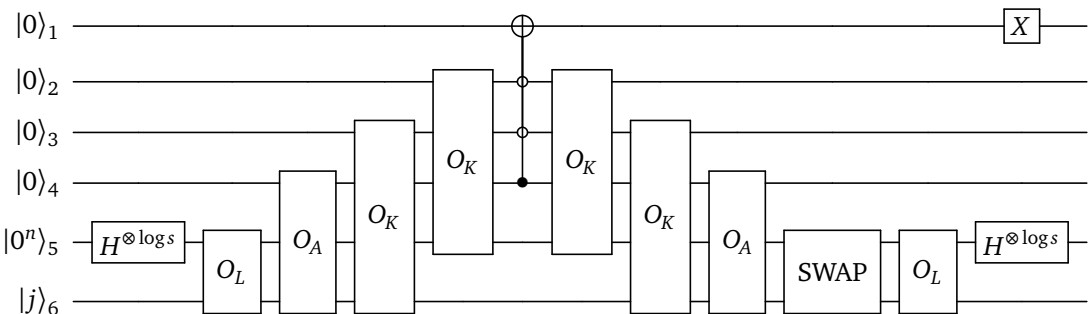

where register "6" has $n$ qubits, the first and fourth $O_K$ are acting on registers "3" and "6", and the second and third $O_K$ are acting on registers 2 and 6.

Repeating a calculation similar to the previous case,

$$|0\rangle_1 |0\rangle_2 |0\rangle_3 |0\rangle_4 |0^n\rangle_5 |j\rangle_6 \xrightarrow{H^{\otimes \log s}} |0\rangle_1 |0\rangle_2 |0\rangle_3 |0\rangle_4 \sum_{l=0}^{s-1} \frac{|l\rangle_5}{\sqrt{s}} |j\rangle_6 \tag{B.15}$$

$$\xrightarrow{O_L} |0\rangle_1 |0\rangle_2 |0\rangle_3 |0\rangle_4 \sum_{l=0}^{s-1} \frac{|r(l,j)\rangle_5}{\sqrt{s}} |j\rangle_6 \tag{B.16}$$

$$\xrightarrow{O_A} |0\rangle_1 |0\rangle_2 |0\rangle_3 \sum_{l=0}^{s-1} \left|A_{r(l,j),j}\right\rangle_4 \frac{|r(l,i)\rangle_5}{\sqrt{s}} |j\rangle_6 \tag{B.17}$$

$$\xrightarrow{O_K \cdot O_K} |0\rangle_1 \sum_{l=0}^{s-1} |r(l,j) \text{ hub?}\rangle_2 |j \text{ hub?}\rangle_3$$

$$\times \left|A_{r(l,j),j}\right\rangle_4 \frac{|r(l,i)\rangle_5}{\sqrt{s}} |j\rangle_6 \tag{B.18}$$

$$\xrightarrow{\text{``Toffoli''}} \sum_{l=0}^{s-1} \left|A_r|_{r(l,j),j}\right\rangle_1 \left|r(l,j)\text{ hub?}\right\rangle_2 \left|j\text{ hub?}\right\rangle_3$$

$$\times \left|A_{r(l,j),j}\right\rangle_4 \frac{\left|r(l,i)\right\rangle_5}{\sqrt{s}} \left|j\right\rangle_6 \tag{B.19}$$

$$\xrightarrow{O_K \cdot O_K} \sum_{l=0}^{s-1} \left|A_r|_{r(l,j),j}\right\rangle_1 \left|0\right\rangle_2 \left|0\right\rangle_3 \left|A_{r(l,j),j}\right\rangle_4 \frac{\left|r(l,i)\right\rangle_5}{\sqrt{s}} \left|j\right\rangle_6 \tag{B.20}$$

$$\xrightarrow{O_A} \sum_{l=0}^{s-1} \left|A_r|_{r(l,j),j}\right\rangle_1 \left|0\right\rangle_2 \left|0\right\rangle_3 \left|0\right\rangle_4 \frac{\left|r(l,i)\right\rangle_5}{\sqrt{s}} \left|j\right\rangle_6 \,. \tag{B.21}$$

The action of the remaining gates on state $\left|0^{n+4}\right\rangle_{1,2,3,4,5} \left|i\right\rangle_6$ is

$$\left|0\right\rangle_1 \left|0\right\rangle_2 \left|0\right\rangle_3 \left|0\right\rangle_4 \left|0^n\right\rangle_5 \left|i\right\rangle_6 \xrightarrow{X} \left|1\right\rangle_1 \left|0\right\rangle_2 \left|0\right\rangle_3 \left|0\right\rangle_4 \left|0^n\right\rangle_5 \left|i\right\rangle_6 \tag{B.22}$$

$$\xrightarrow{H^{\otimes\log s}} \left|1\right\rangle_1 \left|0\right\rangle_2 \left|0\right\rangle_3 \left|0\right\rangle_4 \sum_{l'=0}^{s-1} \frac{\left|l'\right\rangle_5}{\sqrt{s}} \left|i\right\rangle_6 \tag{B.23}$$

$$\xrightarrow{O_H} \left|1\right\rangle_1 \left|0\right\rangle_2 \left|0\right\rangle_3 \left|0\right\rangle_4 \sum_{l'=0}^{s-1} \frac{\left|r(l',i)\right\rangle_5}{\sqrt{s}} \left|i\right\rangle_6 \tag{B.24}$$

$$\xrightarrow{\text{SWAP}} \left|1\right\rangle_1 \left|0\right\rangle_2 \left|0\right\rangle_3 \left|0\right\rangle_4 \sum_{l'=0}^{s-1} \left|i\right\rangle_5 \frac{\left|r(l',i)\right\rangle_6}{\sqrt{s}} \,. \tag{B.25}$$

Then, the inner product gives

$$\left\langle 0^{n+4}\right|_{1,2,3,4,5} \left\langle i\right|_6 U \left|0^{n+4}\right\rangle_{1,2,3,4,5} \left|j\right\rangle_6 = \frac{1}{s} \sum_{l,l'} A_r|_{r(l,j),j} \delta_{i,r(l,j)} \delta_{j,r(l',i)} \tag{B.26}$$

$$= \frac{1}{s} A_r|_{i,j} \,. \tag{B.27}$$

We have used that, if $A_r|_{i,j} \neq 0$, there is a unique $l$ such that $r(l,j) = i$ and that, in that case, there is also a unique $l'$ such that $r(l',i) = j$. If $A_r|_{i,j} = 0$, there are no such $l$ and $l'$, and so (B.26) sums to zero.

- Block-encoding circuit for $A_-$:

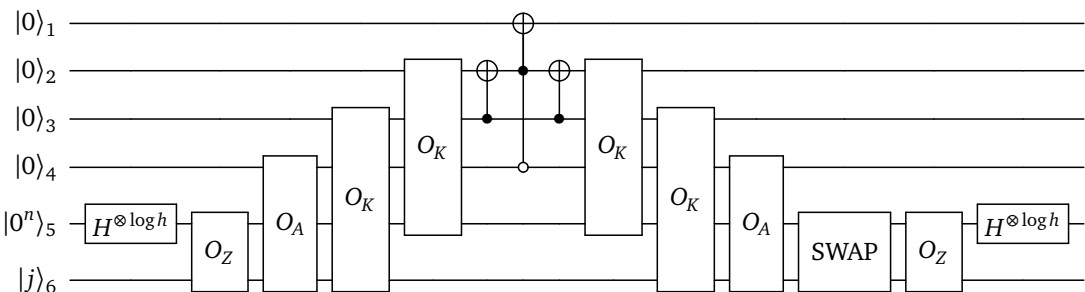

where register "6" has $n$ qubits, the first and fourth $O_K$ are acting on registers "3" and "6", and the second and third $O_K$ are acting on registers 2 and 6.

First,

$$|0\rangle_1 |0\rangle_2 |0\rangle_3 |0\rangle_4 |0^n\rangle_5 |j\rangle_6 \xrightarrow{H^{\otimes \log h}} |0\rangle_1 |0\rangle_2 |0\rangle_3 |0\rangle_4 \sum_{l=0}^{h-1} \frac{|l\rangle_5}{\sqrt{h}} |j\rangle_6 \tag{B.28}$$

$$\xrightarrow{O_Z} |0\rangle_1 |0\rangle_2 |0\rangle_3 |0\rangle_4 \sum_{l=0}^{h-1} \frac{|q(l,j)\rangle_5}{\sqrt{h}} |j\rangle_6 \tag{B.29}$$

$$\xrightarrow{O_A} |0\rangle_1 |0\rangle_2 |0\rangle_3 \sum_{l=0}^{h-1} \left|A_{q(l,j),j}\right\rangle_4 \frac{|q(l,i)\rangle_5}{\sqrt{h}} |j\rangle_6 \tag{B.30}$$

$$\xrightarrow{O_K \cdot O_K} |0\rangle_1 \sum_{l=0}^{h-1} |q(l,j)\text{ hub?}\rangle_2 |j\text{ hub?}\rangle_3$$
$$\times \left|A_{q(l,j),j}\right\rangle_4 \frac{|r(l,i)\rangle_5}{\sqrt{h}} |j\rangle_6 \tag{B.31}$$

$$\xrightarrow{\text{CNOT}} |0\rangle_1 \sum_{l=0}^{h-1} |(j\text{ hub?}) \text{ XOR } (q(l,j)\text{ hub?})\rangle_2 |j\text{ hub?}\rangle_3$$
$$\times \left|A_{q(l,j),j}\right\rangle_4 \frac{|q(l,i)\rangle_5}{\sqrt{h}} |j\rangle_6 \tag{B.32}$$

$$\xrightarrow{\text{``Toffoli''}} \sum_{l=0}^{h-1} \left|A_{-|q(l,j),j}\right\rangle_1 |(j\text{ hub?}) \text{ XOR } (q(l,j)\text{ hub?})\rangle_2$$
$$\times |j\text{ hub?}\rangle_3 \left|A_{q(l,j),j}\right\rangle_4 \frac{|q(l,i)\rangle_5}{\sqrt{h}} |j\rangle_6 \tag{B.33}$$

$$\xrightarrow{\text{CNOT}} \sum_{l=0}^{h-1} \left|A_{-|q(l,j),j}\right\rangle_1 |r(l,j)\text{ hub?}\rangle_2 |j\text{ hub?}\rangle_3$$
$$\times \left|A_{q(l,j),j}\right\rangle_4 \frac{|q(l,i)\rangle_5}{\sqrt{h}} |j\rangle_6 \tag{B.34}$$

$$\xrightarrow{O_K \cdot O_K} \sum_{l=0}^{h-1} \left|A_{-|q(l,j),j}\right\rangle_1 |0\rangle_2 |0\rangle_3 \left|A_{q(l,j),j}\right\rangle_4 \frac{|q(l,i)\rangle_5}{\sqrt{h}} |j\rangle_6 \tag{B.35}$$

$$\xrightarrow{O_A} \sum_{l=0}^{h-1} \left|A_{-|q(l,j),j}\right\rangle_1 |0\rangle_2 |0\rangle_3 |0\rangle_4 \frac{|q(l,i)\rangle_5}{\sqrt{h}} |j\rangle_6 \,. \tag{B.36}$$

Second,

$$|0\rangle_1 |0\rangle_2 |0\rangle_3 |0\rangle_4 |0^n\rangle_5 |i\rangle_6 \xrightarrow{X} |1\rangle_1 |0\rangle_2 |0\rangle_3 |0\rangle_4 |0^n\rangle_5 |i\rangle_6 \tag{B.37}$$

$$\xrightarrow{H^{\otimes \log h}} |1\rangle_1 |0\rangle_2 |0\rangle_3 |0\rangle_4 \sum_{l'=0}^{h-1} \frac{\left|l'\right\rangle_5}{\sqrt{h}} |i\rangle_6 \tag{B.38}$$

$$\xrightarrow{O_Z} |1\rangle_1 |0\rangle_2 |0\rangle_3 |0\rangle_4 \sum_{l'=0}^{h-1} \frac{\left|q(l',i)\right\rangle_5}{\sqrt{h}} |i\rangle_6 \tag{B.39}$$

$$\xrightarrow{\text{SWAP}} |1\rangle_1 |0\rangle_2 |0\rangle_3 |0\rangle_4 \sum_{l'=0}^{h-1} |i\rangle_5 \frac{\left|q(l',i)\right\rangle_6}{\sqrt{h}} \,. \tag{B.40}$$

Finally, if $U$ is the unitary implemented by this circuit,

$$\left\langle 0^{n+4}\right|_{1,2,3,4,5} \langle i|_6 U \left|0^{n+4}\right\rangle_{1,2,3,4,5} |j\rangle_6 = \frac{1}{h} \sum_{l,l'} A_-|_{q(l,j),j} \delta_{i,q(l,j)} \delta_{j,q(l',i)} \qquad \text{(B.41)}$$

$$= \frac{1}{h} A_-|_{i,j}. \qquad \text{(B.42)}$$

We have used that, if $A_-|_{i,j} \neq 0$, there is a unique $l$ such that $q(l,j) = i$ and that, in that case, there is also a unique $l'$ such that $q(l',i) = j$. If $A_-|_{i,j} = 0$, there are no such $l$ and $l'$, and so (B.41) sums to zero.

$\square$

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
