# Peer review of "On the quantum simulation of complex networks"

_SciPost Physics, doi:SciPost Phys. Core 6, 058 (2023)_

## Round 2 · Referee Report · Anonymous · 2023-6-6

Strengths

Nice extension of already known results.

Weaknesses

Too technical.

Report

This manuscript focuses on improved quantum algorithms to deal with
sparse complex network which may have few nodes densely connected
with the rest of the network.

On a technical level, the manuscript considers a very specific case and extend some results,
valid for sparse network, such that, also for slightly more complex network,
one can reduce the complexity of the computation to a poly-logarithmic scaling with the number of nodes.

As far as I understood, the idea is that the adjacency matrix A (Hamiltonian) can be split in such a way that
A = G + (-Am + Ah + Ar), with G easy to exponentiate. Actually G, if collecting all the hubs in the first row(columns), has a very simple structure. Basically spectral decomposition of matrix G is trivial (Appendix A is overcomplicated in my personal opinion).
Actually, the real novelty of the entire proposed approach is based on a trivial decomposition
together with a trivial understanding that exp(-i G t) can be easily prepared
via a series of local quantum operations which properly scale with the number of qubits.
All that summarised in Sec. 5. Then, finally, as already honestly mentioned by the authors,
the problem of solving the time dependent Schroedinger equation in the interaction picture has been
already addressed by Low and Wiebe in [25]. In practice, the only thing that they have still to ensure is an efficient block encoding of the interaction representation of Am(t), Ar(t), and Ah(t).

I would say that this manuscript provides a good description of the
underlying problem, with a reasonable selection of references.
However, from my personal point of view, it is hard to follow for a non specialist.
Although the findings are new and I have generally no complaints concerning the derivations,
I believe that they should be of interest to the experts in the field.

In addition, I have some concerns regarding the assumptions related to the amplitude amplification:
as a generalisation of Grover’s search, I was wandering whether it is not efficient because encoding the classical information to quantum data is slow and makes the overall process inefficient.

Moreover, a still open problem in this approach remain the usual assumption of having access to a series of
“Oracles”; in the specific case OA, OL, OH. In this sense, this could be the mayor limitation of the entire algorithm.

To conclude, I am sorry to say that I cannot argue that these findings, although interesting,
are of a sufficient magnitude, or are likely to be of sufficiently broad interest,
to warrant publication in SciPost Physics.
I would rather suggest that these results would fit better in SciPost Physics Core.

Finally some minor remarks:

1) Between eq (6) and eq(7), there is probably an unwonted repetition “.in the i-th row of A.”

2) In Sec. 6 : “Schödinger” —> “Schrödinger”

---

## Round 2 · Referee Report · Anonymous · 2023-6-24

Report

Overall, I think this is a well-written paper that tackles a fundamental problem and makes modest progress toward it by cleverly finding a solvable subproblem and solving it with a careful combination of known techniques. I recommend acceptance.

The detailed report is attached.

Attachment

---

## Round 3 · List of Changes

1) We removed an unwanted repetition of "in the i-th row of A" 2) We corrected the typo “Schödinger” -> “Schrödinger”

---

## Editorial Decision

published